# Least-Loaded Expert Parallelism: Load Balancing An Imbalanced Mixture-of-Experts

**Xuan-Phi Nguyen** [1]   **Shrey Pandit** [1]   **Austin Xu** [1]   **Caiming Xiong** [1]   **Shafiq Joty** [1]

## Abstract

Mixture-of-Experts (MoE) models are typically pre-trained with explicit load-balancing constraints to ensure statistically balanced expert routing. Despite this, we observe that even well-trained MoE models exhibit significantly imbalanced routing. This behavior is arguably natural—and even desirable—as imbalanced routing allows models to concentrate domain-specific knowledge within a subset of experts. Expert parallelism (EP) is designed to scale MoE models by distributing experts across multiple devices, but with a less-discussed assumption of balanced routing. Under extreme imbalance, EP can funnel a disproportionate number of tokens to a small number of experts, leading to compute- and memory-bound failures on overloaded devices during post-training or inference, where explicit load balancing is often inapplicable. We propose **Least-Loaded Expert Parallelism (LLEP)**, a novel EP algorithm that dynamically reroutes excess tokens and associated expert parameters from overloaded devices to underutilized ones. This ensures that all devices complete their workloads within the minimum collective latency while respecting memory constraints. Across different model scales, LLEP achieves up to $5\times$ speedup and $4\times$ reduction in peak memory usage compared to standard EP. This enables faster and higher-throughput post-training and inference, with $\sim 1.9\times$ faster for gpt-oss-120b. We support our method with extensive theoretical analysis and comprehensive empirical evaluations, including ablation studies. These results illuminate key trade-offs and enable a principled framework for hardware-specific hyperparameter tuning to achieve optimal performance.[2]

[1]Salesforce AI Research. Correspondence to: Xuan-Phi Nguyen <xnguyen@salesforce.com>.

*Proceedings of the $43^{rd}$ International Conference on Machine Learning*, Seoul, South Korea. PMLR 306, 2026. Copyright 2026 by the author(s).

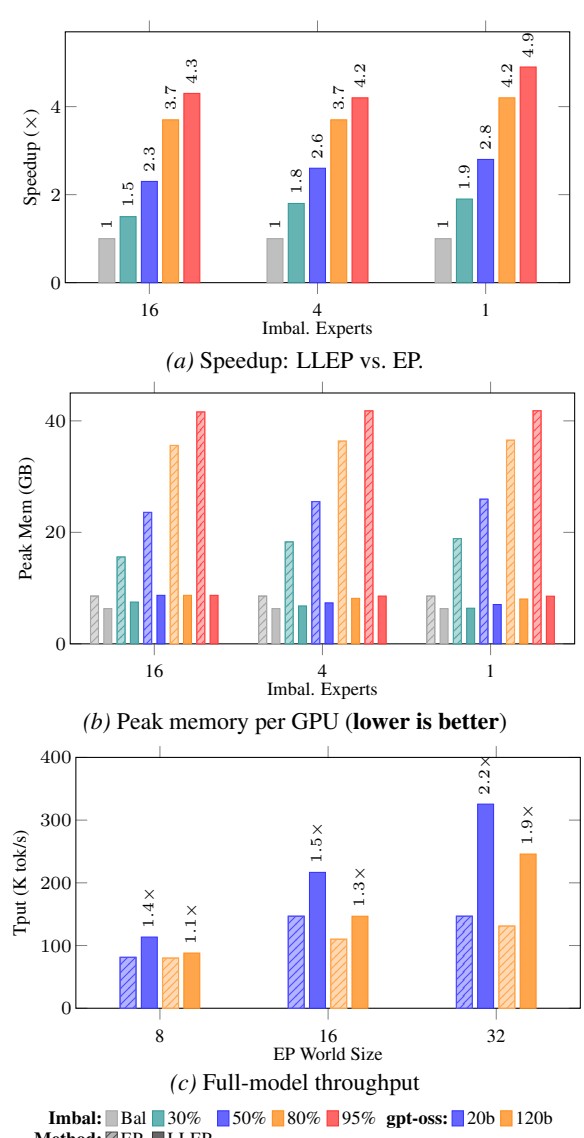

*(a)* Speedup: LLEP vs. EP.

*(b)* Peak memory per GPU (**lower is better**)

*(c)* Full-model throughput

*Figure 1.* LLEP vs. expert parallelism (EP). **(a)** & **(b)** show the speedup and memory of an MoE layer (128 experts, 4 active experts, hidden size of 2048) under perfectly balanced and various imbalanced scenarios: from 30% to 95% of tokens concentrated into 16, 4, 1 imbalanced experts. LLEP is faster than EP by $5\times$ under extremely imbalanced cases, while keeping memory usage stable. **(c)** Realistic full-model throughput: up to $2.2\times$ for gpt-oss-20b and $1.9\times$ for gpt-oss-120b.

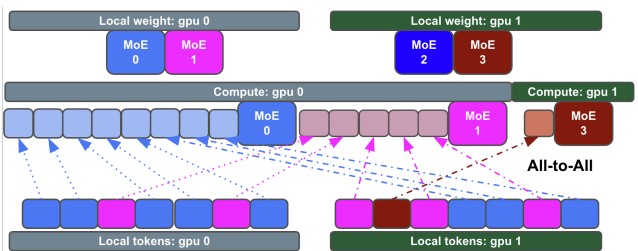
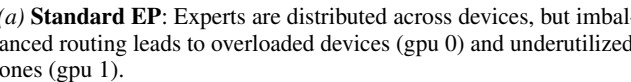

*(a)* **Standard EP**: Experts are distributed across devices, but imbalanced routing leads to overloaded devices (gpu 0) and underutilized ones (gpu 1).

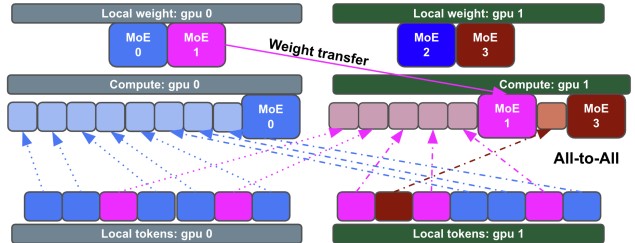

*(b)* **LLEP**: Dynamically redistributes excess tokens and corresponding expert weights from overloaded devices to underloaded devices for balanced execution.

*Figure 2.* Comparison of standard Expert Parallelism and LLEP.

## 1. Introduction

Mixture-of-Experts (MoE) layers have increasingly become indispensable components in large language models (LLMs) due to their ability to scale model size while keeping compute usage per token (activated parameters) constant (Liu et al., 2024; Agarwal et al., 2025; Yang et al., 2025; Team et al., 2025). An MoE module consists of a *router* that selects the top-$K$ experts to route each token to, and a set of feed-forward experts that process the routed tokens. During pre-training, an MoE layer is often externally load-balanced so that statistically diverse batches of tokens are routed evenly to all experts. This is either enforced with an auxiliary loss (Fedus et al., 2022) or stochastic bias terms (Liu et al., 2024). **Expert parallelism (EP)** has become the default infrastructure setup for MoE model training and inference (Singh et al., 2023; Zheng et al., 2024). It spreads the expert weights across multiple (GPU) devices. Under EP, tokens from different devices are *dispatched* to devices that host the experts they are routed to via an All-to-All communication (Shoeybi et al., 2019). Then, each device performs the computation with its local expert weights. The tokens' output are then routed, or *combined*, back to their original devices with another All-to-All (see Fig. 2).

EP is designed with an assumption that load per GPU is always approximately balanced. However, in practice, that is rarely the case. Well-trained MoE models are shown to exhibit consistent imbalanced expert routing (Liu et al., 2024; Agarwal et al., 2025), and arguably for a good reason – as MoE layer training converges, some experts may become specialized to a certain domain or task, while others become generalized across a broad range of knowledge. During domain-specific post-training or inference, only experts relevant to the tasks at hand are mostly activated while others stay dormant. So imbalanced routing is a natural and desirable behavior for MoE models (Qiu et al., 2025). During post-training or inference, parameter-altering load balancing, like auxiliary losses, is discouraged or not allowed to preserve the integrity of the pre-trained MoE routing behaviors (Huang et al., 2024; Hu et al., 2025). Standard EP,

as such, is not designed to handle this phenomenon efficiently. Under worst-case imbalanced scenarios, EP may concentrate an overwhelming number of tokens from all devices to a few overloaded devices. This may cause high computational latency or even out-of-memory (OOM) failures if a GPU cannot store or process the excess tokens. Naive mitigation methods, such as lowering the batch size reduce throughput and increase latency. Advanced strategies like using redundant experts (Liu et al., 2024), meanwhile, increase memory consumption, is only applicable for inference and still fails in the worst cases.

To tackle this problem, we propose **Least-Loaded Expert Parallelism (LLEP)**, a novel EP algorithm that dynamically routes excess tokens, along with their corresponding expert weights, from overloaded devices to underloaded ones. Conceptually, when the globally assigned load on an expert group on a GPU exceeds a capacity threshold, LLEP will only assign tokens to that GPU up to that threshold, and transfer the remaining tokens and the corresponding expert weights to the least-loaded devices for them to "share" the excess workload. LLEP aims to distribut workloads and memory usage across devices such that all devices complete their tasks roughly with the same minimal latency, while maintaining minimal peak memory usage. LLEP is conscious not only about compute load, but also about per-GPU memory allocations and communication overhead. Specifically, an excess tokens transfer is only triggered when the cost of transferring the tokens is less than the cost of processing them locally. Moreover, LLEP comes with backward-pass support, which allows it to be applied dynamically at every iteration of the training loop as well as inference. Importantly, LLEP is an **exact** MoE computation algorithm. Unlike others (Qiu et al., 2025; Hu et al., 2025), it does not alter the models' behaviors for the sake of efficiency.

LLEP demonstrates significant speedup and peak-memory reduction over standard EP. As shown in Figure 1a, it achieves up to 5× speedup under extremely imbalanced scenarios, while maintaining a similar throughput as standard EP when the routing is balanced. Regarding peak memory

usage per GPU, LLEP maintains a relatively stable peak-memory consumption across all scenarios, while standard EP's memory usage grows dramatically with imbalance, up to $4\times$, which will cause OOM crashes if any GPU does not have enough reserve. As such, LLEP is able to handle large models with fewer GPUs while maintaining high throughput. In end-to-end testing with full pre-trained models, LLEP achieves up to $1.4\times$ and $1.9\times$ speedups for gpt-oss-20b and gpt-oss-120b respectively. We conduct a comprehensive theoretical and empirical analysis to provide insights into the cost dynamics of MoEs and expert parallelism, and discuss the trade-offs and design knobs for hardware-specific configuration tuning for best performance.

## 2. Background

### 2.1. Mixture-of-Experts

Since the pioneering work of Lepikhin et al. (2020), MoE models have become the de-facto standard for scaling LLMs (Liu et al., 2024; Agarwal et al., 2025; Team et al., 2025). The MoE layer enables models to learn extensive knowledge across different expert weight matrices, while allowing individually tokens to be processed by a sparse subset of experts, thus improving efficiency and scalability. While concrete implementations vary, MoE architectures rely on the core idea of a *router* layer, which selects the top-$K$ experts to route each token. Tokens are subsequently processed by each activated expert, which can be constructed as a feed-forward (FFN) layer. Formally, given a token $x$'s hidden representation $\boldsymbol{u} \in \mathbb{R}^D$, the MoE module has $N$ experts $\{\text{FFN}_0, \dots, \text{FFN}_{N-1}\}$ and a router layer Router that selects the top-$K$ experts to route $x$ to. For brevity, we define $\text{FFN}_i(\boldsymbol{u}) = \boldsymbol{u}^T \boldsymbol{W}_i$ where $\boldsymbol{W}_i \in \mathbb{R}^{D \times H}$ is the weight matrix of expert $i$, and $\boldsymbol{W}_r \in \mathbb{R}^{D \times N}$ is the weight matrix of the router layer. Then, the MoE output $\boldsymbol{h}$ is

$$\boldsymbol{h} = \sum_{i=0}^{N-1} g_i \text{FFN}_i(\boldsymbol{u}), \quad \text{where} \tag{1}$$

$$g_i = \begin{cases} s_i, & \text{if } s_i \in \text{top-}K(\{s_j \mid 0 \le j \le N-1\}, K) \\ 0, & \text{otherwise} \end{cases} \tag{2}$$

$$s_i = \text{softmax}_i(\boldsymbol{u}^T \boldsymbol{W}_r) \tag{3}$$

**Implementation.** The number of GEneral Matrix Multiplications (GEMMs) required to process one token scales with the number of activated experts. An efficient approach is form per-expert token batches via re-indexing. Assume that for a batch of tokens $\boldsymbol{B} \in \mathbb{R}^{B \times D}$, $G \le N$ experts are activated, i.e., receive routed tokens. Without loss of generality, assume these $G$ activated experts are experts $0, \dots, G-1$. Then, we can form a re-indexed $\boldsymbol{B}' = [\boldsymbol{B}_0, \dots, \boldsymbol{B}_{G-1}]$, where $\boldsymbol{B}_i \in \mathbb{R}^{B_i \times D}$ consists of

---

**Algorithm 1** Highly Efficient Expert Parallelism dispatch_combine: operation per device $p$ (zero-indexed), EP world size $P$, number of experts per device $M = N/P$

**Input:** $K$-repeated input tokens $\mathcal{B}_p \in \mathbb{R}^{B_p \times K \times D}$, router weights $\mathcal{G}_p \in \mathbb{R}^{B_p \times K}$, router indices $\mathcal{I}_p \in \mathbb{Z}^{B_p \times K}$, local expert weights $\boldsymbol{W}_i \in \mathbb{R}^{D \times H}$ for $i = pM, pM + 1, \dots, (p+1)M - 1$

// Dispatch: route tokens and weights to devices that host the experts they are routed to
$\bar{\mathcal{I}}_p \leftarrow \text{sort}(\text{flatten}(\mathcal{I}_p, \text{dims}=[B_p, K]))$
$\bar{\mathcal{B}}_p \leftarrow \text{index\_select}(\text{flatten}(\mathcal{B}_p, \text{dims}=[B_p, K]), \bar{\mathcal{I}}_p)$
$\bar{\mathcal{G}}_p \leftarrow \text{index\_select}(\text{flatten}(\mathcal{G}_p, \text{dims}=[B_p, K]), \bar{\mathcal{I}}_p)$
$\{\boldsymbol{B}_i | i \in \text{unique}(\bar{\mathcal{I}}_p)\} \leftarrow \text{slice}(\bar{\mathcal{B}}_p)$
$\{\boldsymbol{G}_i | i \in \text{unique}(\bar{\mathcal{I}}_p)\} \leftarrow \text{slice}(\bar{\mathcal{G}}_p)$
$\{\hat{\boldsymbol{B}}_i | i \in [pM, (p+1)M - 1]\} \leftarrow \text{All-to-All}(\{\boldsymbol{B}_i\})$
$\{\hat{\boldsymbol{G}}_i | i \in [pM, (p+1)M - 1]\} \leftarrow \text{All-to-All}(\{\boldsymbol{G}_i\})$
// compute Grouped-GEMMs for local experts
$\{\hat{\boldsymbol{H}}_i = \hat{\boldsymbol{G}}_i \odot \hat{\boldsymbol{B}}_i \boldsymbol{W}_i | i \in [pM, (p+1)M - 1]\}$
// Combine: route outputs to their origins
$\{\boldsymbol{H}_i\} \leftarrow \text{All-to-All-reverse}(\{\hat{\boldsymbol{H}}_i\})$
// reverse the sorting and reindexing
$\bar{\mathcal{H}}_p \leftarrow \text{concat}(\{\boldsymbol{H}_i\})$
$\mathcal{H}_p \leftarrow \text{reverse\_sort}(\bar{\mathcal{H}}_p, \bar{\mathcal{I}}_p)$
$\mathcal{H}_p \leftarrow \text{reshape}(\mathcal{H}_p, (B_p, K, H))$
$\mathcal{H}'_p \leftarrow \text{sum}(\mathcal{H}_p, \text{dim=K})$
**Output:** $\mathcal{H}'_p$

---

all tokens routed to expert $i$. For, if $\boldsymbol{B}$ consists of four tokens $\boldsymbol{B} = [\boldsymbol{a}, \boldsymbol{b}, \boldsymbol{c}, \boldsymbol{d}]$ routed to experts $[2, 1, 2, 1]$, then we can form $\boldsymbol{B}' = [\boldsymbol{b}, \boldsymbol{d}, \boldsymbol{a}, \boldsymbol{c}]$ with $\boldsymbol{B}_1 = [\boldsymbol{b}, \boldsymbol{d}], \boldsymbol{B}_2 = [\boldsymbol{a}, \boldsymbol{c}]$. Then, each MoE layer computes $G$ GEMMs $\boldsymbol{B}_i \boldsymbol{W}_i$ for experts $i = 0, \dots, G-1$. The results are then scaled by $g_i$ and summed up according to Eq. (1).

### 2.2. Expert Parallelism

To train MoE models with multiple GPUs, expert parallelism (EP) is generally preferred over tensor or pipeline parallelism (Shoeybi et al., 2019), as it enables more efficient utilization of memory and communication bandwidth. In EP, experts are distributed across GPUs, with each device hosting a local subset of experts. Specifically, input tokens are first processed by a router layer to produce global routing indices and corresponding routing weights (affinity scores). The resulting (input, indices, weight) tuples are then routed to different devices for computation. The routing process is typically conducted using the *dispatch-combine* procedure. Alg. 1 formalizes a highly-efficient per-device implementation of this procedure, while Fig. 2a provides a visual illustration of EP under an imbalanced routing scenario. Specifically, during *dispatch*, each device sends its local input tokens to foreign experts' devices and

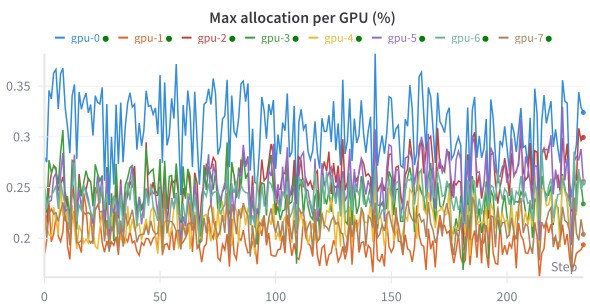

*Figure 3.* Expert routing imbalances across all layers of gpt-oss-20b across batches of data. GPU 0 has 30+% vs. ∼12.5% balanced. Note that load numbers do not add up to 100% because values are maximums across all layers (see Appendix).

receives foreign tokens assigned to its local experts. This data exchange paradigm is called an *All-to-All* communication operation (Sewell et al., 2024; Punniyamurthy et al., 2024). After expert FFN computation is completed, outputs are aggregated in the *combine* stage. Here, all expert outputs are sent back to their originating devices via another All-to-All. Beyond standard NCCL-based collectives, EP can be implemented more efficiently at the kernel level using specialized libraries, such as DeepEP (Liu et al., 2024).

## 3. Analysis

### 3.1. Imbalanced Routing

Even high-performing MoE models have been shown to experience imbalance expert routing, where tokens are routed to a small subset of experts (Liu et al., 2024; Team, 2025). We investigate the dynamics of token-routing by running gpt-oss-20b (Agarwal et al., 2025) through many batches of data, under 8-way EP. To keep the data distribution familiar, we feed the model with conversation data where the questions come from DeepScaleR (Luo et al., 2025) and responses are generated from gpt-oss-20b itself. In Fig. 3, we observe that gpu-0 consistently has the highest load among all GPUs of 30%-35%, more than twice the supposed ∼12.5% balanced load. This indicates that certain GPU devices may handle an overwhelming number of tokens under extremely imbalanced routing. The degree of imbalance changes on a per-batch basis, with no predictability.

**Are imbalanced MoE models actually bad?** It may be tempting to attribute imbalanced routing to pre-training deficiencies, like skewed data or poorly designed load-balancing losses, e.g., Zhou et al. (2022); Shazeer et al. (2017). It is true that if not carefully trained, MoE models can exhibit "expert collapse", where only a small subset of experts is ever activated, producing sub-optimal or weak models. However, as we showed previously, even state-of-the-art MoE models exhibit some degree of imbalanced routing, albeit to a much milder degree than extreme expert collapse.

**Algorithm 2** Least-Loaded Assignment (LLA): Calculate tokens and weights assignment plans. Details *omitted*, see Supplementary for code oracle.

---

**Input:** global expert loads $l \in \mathbb{R}^N$, # local experts $M$, factor $\alpha$, minimum tokens per GEMM $m$
// $\hat{l}$ is sorted loads, $I_{\hat{l}}$ is sorted indices
$\hat{l}, I_{\hat{l}} \leftarrow$ sort($l$, decreasing=true)
// native/pending/assigned load per GPU
$g_n \in \mathbb{Z}^P \leftarrow$ sum of loads of local experts
$g_p \leftarrow g_n$
$g_a \in \mathbb{Z}^P \leftarrow 0$ (zeros)
$m_\alpha \leftarrow \alpha \times \frac{1}{P} \times \sum_{i=0}^{N-1} \hat{l}_i$ //max tokens per GPU allowed
$\mathcal{A} \leftarrow \{\}$ // assignments map for each expert
**for** $i, e$ in zip($I_{\hat{l}}, \hat{l}$) **do**
  $ng \leftarrow$ floor($i/M$)
  $g_p[ng] \leftarrow g_p[ng] - e$
  // available tokens on native GPU
  $na \leftarrow m_\alpha - g_a[ng] - g_p[ng]$
  $A \leftarrow []$ // assignments
  **if** $na \geq e$ or $e - na < m$ **then**
    // Case 1: Native GPU handle all tokens
    $A \leftarrow A + [(ng, 0, e)]$
    $g_a[ng] \leftarrow g_a[ng] + e$
  **else if** $na > 0$ **then**
    // Case 2: Native GPU takes what it can, spill rest
    $nc \leftarrow$ min($na, e$)
    $to \leftarrow nc$ // token offset
    $A \leftarrow A + [(ng, 0, nc)]$
    $g_a[ng] \leftarrow g_a[ng] + nc$
    $r \leftarrow e - nc$ // remaining
    Call LLAS($ng, r, to, A, g_a, g_p, m_\alpha, m$) in Alg. 3
  **else**
    // Case 3: Native GPU overflowed, spill all
    Call LLAS($ng, e, 0, A, g_a, g_p, m_\alpha, m$)
  **end if**
  $\mathcal{A}[i] \leftarrow A$
**end for**
$\mathcal{W} \leftarrow$ construct weight transfer plan from $\mathcal{A}$
**Output:** $\mathcal{A}, \mathcal{W}$

---

Rather than aiming for perfectly balanced routing, we consider mild imbalance a natural property of a well-trained MoE model. After undergoing large-scale pre-training, subsets of experts often specialize in particular knowledge domains, tasks, or capabilities (Qiu et al., 2025; Hu et al., 2025; Song et al., 2025). Consequently, when an MoE model is further fine-tuned or evaluated on a specific domain, such as mathematics, experts specialized for that domain are activated more frequently. This leads to imbalanced routing. At the same time, some experts may evolve into broadly applicable, domain-agnostic "shared" experts that consistently handle generic linguistic patterns, such as grammar, across tasks. This phenomenon has also been observed and

embraced in prior work (Liu et al., 2024). From this perspective, aggressively enforcing balanced routing, e.g., by altering model behavior through auxiliary load-balancing losses (Fedus et al., 2022) or moving-average routing biases (Liu et al., 2024), risks disrupting these learned specialization patterns within particular experts. Instead of correcting imbalance at the model level, we instead embrace it, proposing a *system-level* mechanism for both training and inference for maximizing throughput under imbalanced routing. This respects the inherent specialization among experts while mitigating inefficiencies that arise with imbalance.

Several approaches have been proposed to mitigate routing imbalance under EP. A naive solution is to reduce the batch size, but this severely degrades throughput. Another strategy employs chained gradient checkpointing to process tokens in smaller chunks; however, this remains inefficient and is still constrained by a hard memory ceiling. For inference, Liu et al. (2024) propose an EP Load Balancer (EPLB) that replicates heavily loaded experts across devices; or Doucet et al. (2025) suggest an asynchronous weight prefetch scheme. While effective in some settings, these methods may incur additional memory overhead, are not applicable for training; further, this can still result in out-of-memory (OOM) failures under extreme routing imbalance. Huang et al. (2024) suggest reserving additional memory for excess experts, which likewise incurs CPU and GPU memory overhead.

### 3.2. Distributed Latency and Memory Analysis

To gain insight into the worst-case cost model of MoE layers under EP, we analyze both latency and peak memory usage in a holistic manner. Here, we consider the computation local to a single GPU device, while the communication overheads are discussed in the Appendix. Given a batch routed to $G$ local experts, the MoE layer performs $G$ GEMMs, and the total latency can be approximated as

$$T_{\text{local}} = \sum_{i=0}^{G-1} (T_{\text{overhead}} + B_i \times T_{B_i,D,H}) \qquad (4)$$

where $T_{\text{overhead}}$ denotes the kernel launch latency, and $T_{B_i,D,H}$ is the per-token compute time, which depends on the token count $B_i$ and model dimensions $D$ and $H$ (defined in § 2.1). The efficiency of $T_{B_i,D,H}$ is directly impacted by how GEMM kernels are implemented, optimized, and tuned with respect to different input and output sizes and configurations. In general, GEMMs become more efficient as $B_i$, $D$ and $H$ increase. For example, with $D$ and $H$ fixed, $T_{B_1,D,H} < T_{B_2,D,H}$ when $B_1 > B_2$. Therefore, given a fixed number of FLOPs, executing a small number of large GEMMs is significantly more efficient than executing many small GEMMs. EP exploits this property by aggregating tokens across devices, thereby reducing the number of local experts $G$ and increasing the effective batch size $B_i$ per

---

**Algorithm 3** Least-Loaded Assignment Spill (LLAS): Spilling the remaining tokens to other GPUs

**Input:** native GPU $ng$, remaining tokens $r$, offset $to$, assignments $A$, assigned load $g_a$, pending load $g_p$, $m_\alpha$, $m$
**while** $r > 0$ **do**
  $o \in \mathbb{Z}^{P-1} \leftarrow$ other GPUs $g \neq ng$ sorted by $g_a[g] + g_p[g]$
  **for** $o$ in $o$ **do**
    $c \leftarrow \min(r, m_\alpha - g_a[o] - g_p[o])$
    **if** $c < m$ and $r > c$ **then**
      skip // chunk too small
    **end if**
    $A \leftarrow A + [(o, to, to + c)]$ // assign load
    $g_a[o] \leftarrow g_a[o] + c$
    $r \leftarrow r - c$
    $to \leftarrow to + c$ // increment token offset
    break
  **end for**
  **if** none of $o$ assigned **then**
    $o \leftarrow o[0]$ //force assign the least loaded GPU
    $A \leftarrow A + [(o, to, to + r)]$
    $g_a[o] \leftarrow g_a[o] + r$
    $r \leftarrow 0$
  **end if**
**end while**

---

expert. Fig. 8 in the Appendix provides further analysis on this. The peak memory usage of the MoE layer is defined approximately as:

$$M_{\text{local}} = \sum_{i=0}^{G-1} (B_i \times D + D \times H + B_i \times H) \qquad (5)$$

Under standard expert parallelism, $B_i$ is the total number of tokens routed to expert $i$ from across all EP devices. In the worst case, $B_i$ may approach the global batch size, causing all tokens to be concentrated on a single device while others are idle. This causes spiking latency and memory usage, or even out-of-memory crashes for the overloaded device. Figs. 1a and 1b show the slowdown and peak memory usage of a standard EP setup under different imbalance scenarios. As shown in Fig. 1a, EP could be 4.6x slower when 95% of tokens are routed to a single expert compared to the balanced baseline. Meanwhile, EP's peak memory usage per GPU may grow up to 4x, potentially causing OOM errors.

## 4. Least-Loaded Expert Parallelism (LLEP)

We explain in detail how our proposed LLEP works. Conceptually, our method will detect ahead of time the degree of imbalance of the global routing according to per-expert loads. If the imbalance is lower than a threshold $\lambda$, then we

consider the routing as balanced and proceed to the standard EP procedure. Otherwise, we will execute the least-loaded assignment algorithm (Alg. 2) to determine for each GPU device that it needs to compute GEMMs for which experts and with how much portions of the global tokens routed to them. If the GPU does not contain an assigned expert as resident, it will import the expert from its host GPU. The assignment takes into account the overhead cost of weight and data transfers, in comparison to the latency and memory cost of processing the tokens only for local experts. Algs. 2 to 4 formally describe our method in detail.

**Constraints.** LLEP works by making routing decisions that are subject to some constraints. First, factor $\alpha$ in Alg. 2 determines how much maximum token capacity a GPU can handle, which we defined as $m_\alpha = \alpha \sum_{i=0}^{N-1} \boldsymbol{l}_i / P$ tokens. $m_\alpha$ is not necessarily a physical memory limit, but rather a threshold that the GPU is considered overloaded. If a local expert load exceeds $m_\alpha$, it will spill the excess load to other GPUs. Second, $m$ is the minimum tokens per GEMM for it to be efficient. If a local expert load exceed the local GPU's occupied capacity, but the excess is less than $m$, we consider it's not worth it to spill and instead force the local GPU to compute it despite over-capacity (see § 3.2). Third, imbalance ratio threshold $\lambda$ is used to determine whether the global loads are relatively balanced, in which case we switch back to standard EP. The reason is that our method employs a greedy least-loaded assignment (LLA) algorithm (Alg. 2) that would produce the same routing plan as standard EP anyway, while causing a tiny time overhead. Without skipping this imbalance ratio check, our method is shown to be slightly slower than standard EP under perfect balance. The values of $\alpha$, $m$, and $\lambda$ determine the threshold for when the cost of local computation exceeds that of data transfers. The optimal values for them depend on $N$, $P$, $B_p$, $K$, $D$, $H$, the model size, and the physical system. Thus, we recommend to tuning them for each use case.

**Elaboration.** The least-loaded assignment (LLA) algorithm (Alg. 2) determines, for each expert, which GPUs handle which portions of the global expert's load. First, it sorts the expert loads in decreasing order. Then, it determines the GPU allocations for each expert from largest-load to smallest-load ones. For each expert, it first determines if the native GPU (the one that hosts the expert's weights) can handle all the tokens of the expert. If it can, it assigns all the tokens to the native GPU. If it cannot, it spills the excess tokens to the least-loaded available GPU up to the capacity threshold. If there are still remaining excess tokens, it will continue this spilling loop (LLAS, Alg. 3) until all the tokens are assigned. Once the tokens routing plan is finalized, it will also construct the weight transfer plan accordingly. For example, if excess load of expert $i$ native to GPU $p$ is spilled to GPU $q$, then the weight transfer plan will

---

**Algorithm 4** LLEP dispatch_combine: operation per device $p$ (zero-indexed), EP world size $P$, number of experts per device $M = N/P$

**Input:** $\mathcal{B}_p \in \mathbb{R}^{B_p \times K \times D}$, $\mathcal{G}_p \in \mathbb{R}^{B_p \times K}$, $\mathcal{I}_p \in \mathbb{Z}^{B_p \times K}$, $\boldsymbol{W}_i \in \mathbb{R}^{D \times H}$ for $i = pM, pM+1, \ldots, (p+1)M - 1$
$\boldsymbol{l} \leftarrow$ sum of loads of global experts across all GPUs
**if** $\max(\boldsymbol{l})/\text{mean}(\boldsymbol{l}) < \lambda$ **then**
    Call standard EP Alg. 1
    **Output:** $\mathcal{H}'_p$ from standard EP
**end if**
$\bar{\mathcal{I}}_p \leftarrow \text{sort}(\text{flatten}(\mathcal{I}_p, \text{dims}=[B_p, K]))$
$\bar{\mathcal{B}}_p \leftarrow \text{index\_select}(\text{flatten}(\mathcal{B}_p, \text{dims}=[B_p, K]), \bar{\mathcal{I}}_p)$
$\bar{\mathcal{G}}_p \leftarrow \text{index\_select}(\text{flatten}(\mathcal{G}_p, \text{dims}=[B_p, K]), \bar{\mathcal{I}}_p)$
// construct routing plan and weight transfer plan
$\mathcal{A}, \mathcal{W} \leftarrow \text{LLA}(\boldsymbol{l}, M)$
$\{\boldsymbol{B}_i | i \in [0, ..., P]\} \leftarrow$ build chunks of $\bar{\mathcal{B}}_p$ from $\mathcal{A}$
$\{\boldsymbol{G}_i | i \in [0, ..., P]\} \leftarrow$ build chunks of $\bar{\mathcal{G}}_p$ from $\mathcal{A}$
// S is expert IDs of foreign experts assigned to this device
$\{\hat{\boldsymbol{B}}_i | i \in [pM, (p+1)M - 1] \cup S\} \leftarrow \text{All-to-All}(\{\boldsymbol{B}_i\})$
$\{\hat{\boldsymbol{G}}_i | i \in [pM, (p+1)M - 1] \cup S\} \leftarrow \text{All-to-All}(\{\boldsymbol{G}_i\})$
$\{\boldsymbol{W}_j | j \in S\} \leftarrow$ P2P Transfer weights from other GPUs to this GPU
// compute GEMMs for native and foreign experts
$\{\hat{\boldsymbol{H}}_i = \hat{\boldsymbol{G}}_i \odot \hat{\boldsymbol{B}}_i \boldsymbol{W}_i | i \in [pM, (p+1)M - 1] \cup S\}$
// Combine: route outputs to their origins
$\{\boldsymbol{H}_i\} \leftarrow \text{All-to-All-reverse}(\{\hat{\boldsymbol{H}}_i\})$
// reverse the sorting and reindexing
$\bar{\mathcal{H}}_p \leftarrow \text{concat}(\{\boldsymbol{H}_i\})$
$\mathcal{H}_p \leftarrow \text{reverse\_sort}(\bar{\mathcal{H}}_p, \bar{\mathcal{I}}_p)$
$\mathcal{H}_p \leftarrow \text{reshape}(\mathcal{H}_p, (B_p, K, H))$
$\mathcal{H}'_p \leftarrow \text{sum}(\mathcal{H}_p, \text{dim=K})$
**Output:** $\mathcal{H}'_p$

---

include a weight transfer operation from $p \rightarrow q$ for $W_i$. The LLA algorithm ensures that each GPU prioritize computing most, if not all, of its local experts' load first before accepting foreign experts' load. This is to minimize the number of weight transfers required. The final LLEP algorithm (Alg. 4) will then execute the dispatch-compute-combine operations according to the routing plans obtained from LLA. Specifically, for each device, in addition to GEMM computation for native experts, LLEP will also compute the GEMMs for foreign experts that are assigned to the device. Fig. 11 (Appendix § C.4) visualizes an example of how LLEP works. Unlike others, LLEP supports proper gradient propagation. During the backward pass, the gradients for the spilled expert weights are returned to their native devices and accumulated with their native gradients respectively.

**Implementation & Optimization.** In the experiments, we implement LLEP with the Torch's NCCL for All-to-All and peer-to-peer (P2P) operatives. While our simple im-

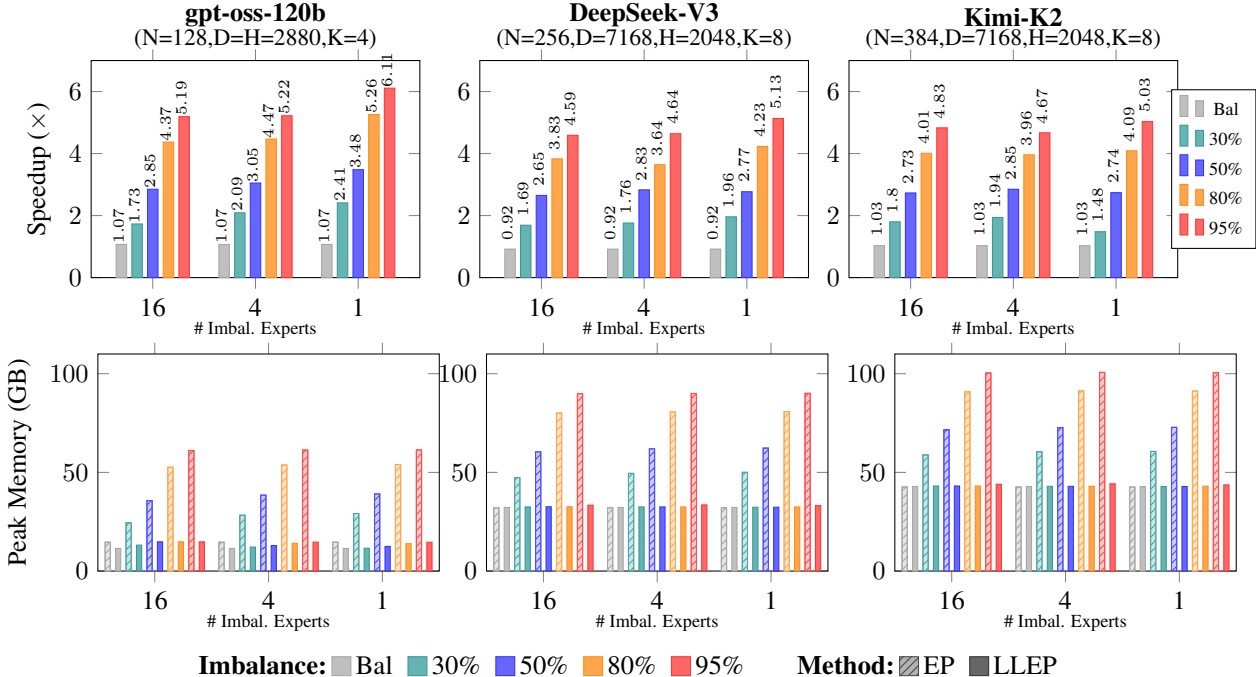

*Figure 4.* Comparison of LLEP vs. standard EP across three MoE architectures. **Top row:** Speedup (×, higher is better) of LLEP over EP. Gray bars show the balanced baseline. Colored bars indicate imbalance levels (percentage of tokens routed to that many experts). Higher concentration yields greater speedup, up to 6.1× for gpt-oss-120b. **Bottom row:** Peak memory usage per GPU (GB, **lower is better**). EP memory grows dramatically with imbalance, while LLEP maintains near-constant memory across all scenarios.

plementation is already showing significant speedup and memory saving, there are further opportunities to optimize. For instance, the communication operatives can be written as low-level C++/Triton kernels, or using a modified version of DeepEP (Liu et al., 2024). Such a fused operative may also perform direct All-to-All on unsorted tensors $\mathcal{B}_p$ and $\mathcal{G}_p$, avoiding the memory-intensive index_select operation (Alg. 4). The communication can be overlapped with computation. For multi-node setups, we can further modify LLEP to prefer spilling work to intra-node devices to limit the higher inter-node communication overhead.

# 5. Experiments

We show the advantages of LLEP under two settings: Controlled experiments, where we precisely simulate imbalanced loads, and end-to-end full-model computations. We conclude with an ablation study, characterizing various hyperparameters. See Appendix for experimental details and metric definitions.

## 5.1. Speed and Memory Profiles

We analyze the speed and memory profiles of popular MoE layers of different configurations, namely those used in gpt-oss-120b (Agarwal et al., 2025), DeepSeek-V3 (Liu et al., 2024) and Kimi-K2 (Team et al., 2025). We benchmark

the forward pass speedups and peak memory consumption per GPU. We simulate across different balanced and imbalanced routing scenarios, from 30% to 95% of tokens evenly concentrated into 1, 4 or 16 experts. For LLEP, we use $\lambda = 1.3, \alpha = 1, m = 1024$. Fig. 4 summarizes the results. As shown in the speedup row, across different configurations, LLEP outperforms standard EP across all imbalance scenarios, achieving greater speedup under more imbalanced routing, up to 6.11× for the most extreme case (95% into 1 expert). Meanwhile, LLEP roughly maintains EP's efficiency in the perfectly balanced case, thanks to the adaptive ratio $\lambda$. In the peak memory row, LLEP maintains consistently and stably low memory consumption across all imbalance scenarios, with memory saving of up to 5×, allowing us to increase the throughput (batch size) without running into out-of-memory (OOM) failure.

## 5.2. End-to-End Full Model Speed Profiles In The Wild

To measure the effectiveness of LLEP in the **wild**, instead of just simulating imbalances, we conduct end-to-end forward-pass throughput comparisons with real pre-trained gpt-oss-20b and gpt-oss-120b (Agarwal et al., 2025) on samples from the Nemotron-Math dataset (Du et al., 2025), where the responses were generated by gpt-oss-120b itself. The results are reported in Fig. 1c. Full model throughput is impacted by other irrelevant factors and fixed overheads,

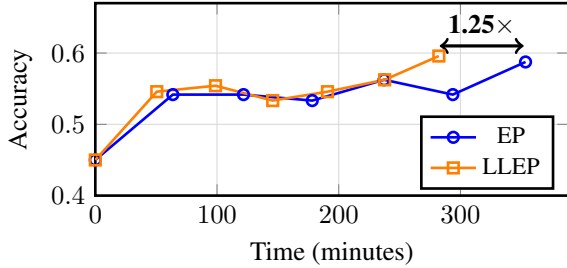

*Figure 5.* Accuracy on AIME'25 vs. wall-time for EP and LLEP when training gpt-oss-20b (low) on full parameters using Zero-3 and CPU offloading for gradients and optimizer states.

such as the attention layers. Thus, the speedup of the MoE layers is always greater than the reported numbers for the full model throughput. As shown, LLEP achieves up to $2.2\times$ and $1.88\times$ speedups for gpt-oss-20b and gpt-oss-120b respectively. Our method achieves better scaling efficiency with greater relative speedups the more GPUs are used. LLEP demonstrates its superiority because the models exhibit imbalanced routing naturally even on in-domain data.

Fig. 5 shows the downstream performance (accuracy on AIME'25) vs. wall-clock time for EP and LLEP, when training gpt-oss-20b on **full** parameters, using Zero-3 and CPU offloading for gradients and optimizer states. The training process requires more expensive and non-negotiable, but irrelevant, overheads, including on-CPU computations of gradients and parameter updates as well as checkpoint saving at each step. As such, LLEP achieves $1.25\times$ speedup over EP while achieving comparable accuracy.

### 5.3. Ablation Study

**Batch size** $B$    Fig. 6a shows that LLEP achieves greater speedups the more tokens we pack into the batch, across various scenarios. The reason is that large batches saturate the capacity of each individual GPU and overwhelm any overhead introduced by Alg. 2, leading to a linear relationship between batch size and processing time. This means that the least collective processing time ($\max_i[\text{time-of-GPU } i]$) is achieved when compute workloads are evenly distributed across all GPUs. The All-to-All data transfers of large batches also overshadow any associated weight transfer.

**Factor** $\alpha$    Fig. 6b shows the speedups across different $\alpha$ values. We observe that higher $\alpha$ leads the lower speedup, meaning allowing more per-GPU capacity before LLA spilling may cause inefficiency. This means that at large-enough batch sizes, LLEP prefers workloads to be balanced despite possibly higher communication costs.

**Adaptive ratio** $\lambda$    Fig. 7a shows how the adaptive ratio $\lambda$ impacts speedup. Specifically, when the batch size is

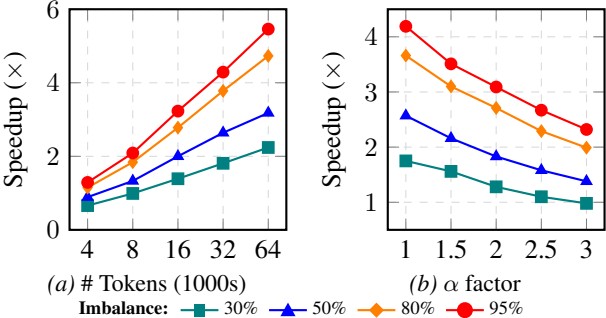

*(a) # Tokens (1000s)*        *(b) $\alpha$ factor*

**Imbalance:** ■ 30%  ▲ 50%  ◆ 80%  ● 95%

*Figure 6.* Speedup of LLEP over standard EP. (a) Speedup as a function of batch size. (b) Speedup as a function of $\alpha$; lower $\alpha$ yields higher speedups. See Appendix § B.5 for configurations.

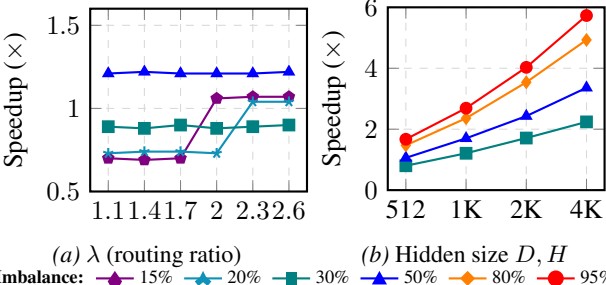

*(a) $\lambda$ (routing ratio)*        *(b) Hidden size $D, H$*

**Imbalance:** ◆ 15%  ✳ 20%  ■ 30%  ▲ 50%  ◆ 80%  ● 95%

*Figure 7.* Speedup of LLEP over EP. (a) Speedup vs. $\lambda$. (b) Speedup scales with hidden size. See Appendix § B.5.

low ($B = 8$K), we observe higher $\lambda$ is beneficial when the imbalance degree is low (15-20%). In other words, it is better to revert to standard EP when the routing distribution is balanced enough that the overhead costs of LLEP's weight transfers surpassed the benefits of even computation.

**Hidden size** $D$ **and** $H$    Fig. 7b shows that LLEP shines as the model's hidden size scales, despite that the weight transfers cost more. The reason for this scaling effect is that as the hidden size increases, the compute efficiency of each GEMM improves compared to the data communication costs. Plus, similar to scaling batch sizes, large hidden sizes also saturate the GPU capacity. This causes the benefits of the compute workloads being perfectly balanced to overshadow any inconvenient weight transfer overhead.

**Larger-scale pre-training**    To further examine scalability, we train a DeepSeek-V3-scale ($\sim$600B) model from scratch using the same load-balancing bias dropless strategy of Liu et al. (2024), which natively mitigates routing imbalance through semantic means. However, randomly-initialized models are extremely imbalanced, and this semantic technique is insufficient to achieve physical load balance. LLEP, by contrast, enforces perfect physical balance across GPUs while preserving mathematical equivalence of the routing, yielding $\sim$1.55$\times$ speedup over standard EP with

load-balancing. EPLB does not support this pre-training setting. These gains exceed our reported $1.25\times$ on gpt-oss-120b fine-tuning, consistent with the observation that higher token imbalance amplifies the benefits of LLEP.

**Inference Analysis** We evaluate LLEP in the inference setting on gpt-oss-120b with 32 GPUs on the DeepScaleR dataset. For the prefill stage, LLEP achieves $1.94\times$ throughput over standard EP, outperforming EPLB (Liu et al., 2024) at $1.45\times$. For the decode stage, LLEP's weight transfer mechanism only activates when the batch size per GPU is significantly larger than the expert weight dimension, so that compute savings from balanced workloads outweigh the transfer overhead; consistent with Fig. 6a, larger batch sizes yield greater speedups. When this condition is met, LLEP achieves $\sim 1.8\times$ speedup over standard EP; otherwise it degrades to standard EP. This makes LLEP particularly suited to tasks with short prompt lengths, where sequences are lightweight and large batch sizes are more readily achievable. Under low batch size regimes where this condition is not met, combining LLEP with EPLB—which replicates hot experts statically—offers a complementary path to balanced decode throughput.

## 6. Conclusion

We present least-loaded expert parallelism (LLEP), a novel EP algorithm that dynamically performs load balancing to address the MoE imbalanced routing phenomenon, while ensuring the exact mathematical computation. LLEP achieves up to 5-6$\times$ speedups and $5\times$ reduction in peak memory consumption for the MoE layers. It improves the end-to-end full-model throughputs of gpt-oss-120b by up to 90%.

## Impact Statement

This paper introduces a new algorithm to compute mixture-of-experts (MoE) computation across multiple compute accelerators, such as GPUs, more efficiently. The algorithm addresses the inefficiency of the expert parallelism baseline when an MoE model exhibits imbalanced routing, which is often inevitable but also rather desirable. By achieving better load balancing, our method achieves higher throughput, faster training and inference, while consuming less memory. As a result, our method saves costs in equipment investments as well as energy resources while achieving the same degrees of model intelligence and capability. Our method helps improve the environment impacts and carbon footprints of AI models.

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

# A. Cost Analysis of Cross-Device Communication

With standard expert parallelism, the communication cost stems from the two 3 All-to-All operations in the MoE layer. Under routing imbalance, the communication cost is also imbalanced. For the dispatch All-to-All, we can split the cost into two parts: $c_{\text{send}}^{\text{dispatch}}$ for sending data and $c_{\text{recv}}^{\text{dispatch}}$ for receiving data.

$$c_{\text{send}}^{\text{dispatch}} = \sum_{i=0}^{N-1} \beta_{\text{send}} \cdot \hat{B}_i \times (D+1) \quad \text{where } i \text{ is not local expert} \tag{6}$$

$$c_{\text{recv}}^{\text{dispatch}} = \sum_{i=0}^{G-1} \beta_{\text{recv}} \cdot (\overline{B}_i - \hat{B}_i) \times (D+1) \quad \text{where } i \text{ is local expert} \tag{7}$$

where $\overline{B}_i$ is the total global number of tokens routed to expert $i$, of which $\hat{B}_i$ is the number of tokens routed to expert $i$ on the local GPU and is not needed to be transferred. The last dimension $(D+1)$ accounts for the input vector and gating score. For the combine All-to-All, the traffic flows in the opposite direction:

$$c_{\text{send}}^{\text{combine}} = \sum_{i=0}^{G-1} \beta_{\text{send}} \cdot (\overline{B}_i - \hat{B}_i) \times H \quad \text{where } i \text{ is local expert} \tag{8}$$

$$c_{\text{recv}}^{\text{combine}} = \sum_{i=0}^{N-1} \beta_{\text{recv}} \cdot \hat{B}_i \times H \quad \text{where } i \text{ is not local expert} \tag{9}$$

Each GPU device has its own bandwidth, these costs consumes the bandwidth. If the bandwidth is saturated, the latency for communication linearly increases with the maximum of the two costs over all GPUs.

With LLEP, there is an additional communication cost for transferring the expert weights, which are Peer-to-Peer communications. The cost for each weight transfer is:

$$c_{\text{send}}^{\text{weight}} = \beta_{\text{send}} \cdot D \times H \tag{10}$$

$$c_{\text{recv}}^{\text{weight}} = \beta_{\text{recv}} \cdot D \times H \tag{11}$$

LLEP is designed to limit the number of P2P transfers as much as possible. Our method will outperforms EP when the least number of P2P transfers is over-compensated for balancing out the 4 All-to-All costs defined above.

# B. Experimental Settings

## B.1. How Speedup and Memory Usage is Measured?

For speedup, we first run the MoE layer (both EP and LLEP) for $n_{\text{warmup}} = 10$ steps to warm up the GPU, when necessary initialization overheads are incurred and stabilized. These initial overheads are one-off and do not represent the true iterative time. Then, we run it again for $n_{\text{measure}} = 100$ steps to measure the total elapsed time since, which the speedup can be derived from. For memory usage, we first clean up all cache memory, and then measure the peak memory usage per GPU after running the MoE layer.

## B.2. How Routing Imbalance is Simulated

We model imbalance via the percentage of global tokens $p\%$ that are evenly concentrated into a subset $C$ of global experts. The choice of $C$ is uniformly randomized across runs. Among selected experts in $C$, each expert get $p/C\%$ of the global tokens, which are also randomly chosen. Among the remaining experts, each expert get $(1-p)/(N-C)\%$ of the global tokens.

## B.3. Hardware

We use AWS hyperpod H200 GPUs with NVLink and Elastic Fabric Adapter (EFA) for communication and InfiniBand topology.

## B.4. Speed & Memory and End-to-End Profiles

We provide more details for the experiments in § 5.1. For the MoE layer structure, unlike the simple case stated in § 2, each MoE expert is a variant of SwigGLU (Shazeer, 2020) feed-forward module that use three weight matrices instead of one. More precisely, each MoE expert consists of $\boldsymbol{W}_i^g, \boldsymbol{W}_i^u \in \mathbb{R}^{D \times H}$ and $\boldsymbol{W}_i^d \in \mathbb{R}^{H \times D}$. Given input token $\boldsymbol{u}$, the expert FFN output is $F\hat{F}N_i(\boldsymbol{u}) = [(\boldsymbol{u}^T \boldsymbol{W}_i^u + 1) \cdot \hat{\boldsymbol{g}}]\boldsymbol{W}_i^d$ with $\hat{\boldsymbol{g}} = \boldsymbol{g} \cdot \sigma(\boldsymbol{g})$ and $\boldsymbol{g} = \boldsymbol{u}^T \boldsymbol{W}_i^g$. We benchmark the forward pass speedups and peak memory consumption per GPU across $P = 8$ H200 GPUs. The batch size per GPU ($B$) is 32K for gpt-oss and 16K for DeepSeek-V3 and Kimi-K2. We simulate across different balanced and imbalanced routing scenarios, from 30% to 95% of tokens evenly concentrated into 1, 4 or 16 experts. For LLEP, we use $\lambda = 1.3, \alpha = 1, m = 1024$.

For the end-to-end full model speedup experiments in § 5.2, we implement the gpt-oss models using a hybrid of Zero-3 parameter sharding and expert parallelism. Specifically, we use Zero-3 for non-MoE parameters and EP parameter sharding for the MoE layers. During forward pass, Zero-3 will all-gather the non-expert parameters when they are needed. Meanwhile, EP will not all-gather the expert parameters – because they are already sharded across EP ranks.

We use a highly optimized implementation of the MoE layer that is **2×** faster than Huggingface naive implementation. We use the same implementation for EP and LLEP for a fair comparison. We use CPU offloading for gradients and optimizer states. Therefore, the irrelevant overheads of CPU transfers and computations, as well as intermittent checkpoint saving, causes significant delays that eat into the speedup of the MoE layers.

As for the training data, we collect traces at low and medium reasoning efforts from the Nemotron-Math dataset (Du et al., 2025), whose samples are generated by gpt-oss-120b itself. We use the data to distill gpt-oss-20b model according to the end-to-end full model training experiment.

## B.5. Experimental Configurations

Table 1 summarizes the configurations used for our ablation experiments in § 5.3.

*Table 1.* Experimental configurations for ablation studies. The "Varying" column indicates which parameter is swept.

| Varying | $B$ | $D$ | $H$ | $N$ | $k$ | $\alpha$ | $m$ | $\lambda$ |
|---|---|---|---|---|---|---|---|---|
| $B$ (Fig. 6a) | 4K–64K | 2048 | 2048 | 128 | 4 | 1.0 | 1024 | 1.3 |
| $\alpha$ (Fig. 6b) | 32K | 2048 | 2048 | 128 | 4 | 1–3 | 1024 | 1.3 |
| $\lambda$ (Fig. 7a) | 8K | 2048 | 2048 | 128 | 4 | 2.0 | 1024 | 1.1–2.6 |
| $D, H$ (Fig. 7b) | 32K | 512–4K | 512–4K | 128 | 4 | 1.0 | 1024 | 1.3 |

# C. Additional Results

## C.1. Separate vs Fused Grouped-GEMM

Fig. 8 shows the compute time between a naive for-loop of GEMMs using cuBLAS implementation vs a fused optimized Grouped-GEMM kernel written in Triton, with adoption of Tensor Memory Accelerator (TMA). The cuBLAS version launches $N$ GPU kernel launches, causing high overhead, while the Triton version launches only one. However, as shown, the cuBLAS version still outperforms Triton counterpart because each cuBLAS GEMM kernel is hardware-specific and highly optimized at architecture level, while the Triton version is a generic implementation. In addition, despite all computations have the same FLOPs, the compute time dramatically increases the more experts are present. Therefore, it is better to compute a few giant GEMMs with few experts than to compute many tiny GEMMs with many experts. Both expert parallelism and our method (LLEP) leverage this principle by spreading expert weights across EP ranks, allowing each rank to compute only a handful of experts.

## C.2. Imbalanced Routing

We investigate the dynamics of token-routing by running gpt-oss-20b (Agarwal et al., 2025) through many batches of data, under 8-way EP. To keep the data distribution familiar, we feed the model with conversation data where the questions come from DeepScaleR (Luo et al., 2025) and response chain-of-thoughts are generated from gpt-oss-20b itself. We visualize the maximum load per expert and per GPU in Fig. 9.

In Fig. 9a, we observe that tokens are consistently routed to certain expert positions, with position E11 dominating. This

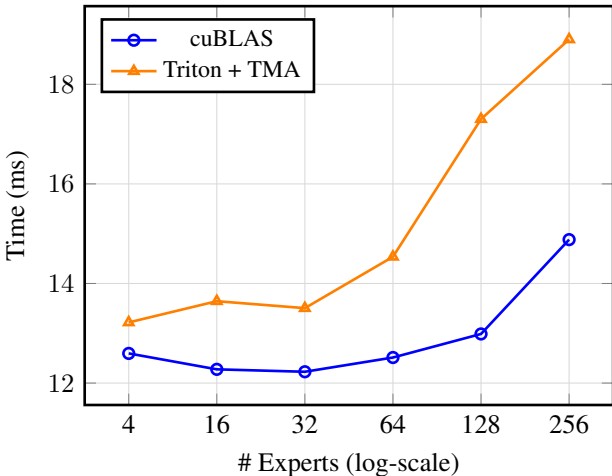

*Figure 8.* Grouped-GEMM benchmark (**lower is better**): execution time vs. number of experts under balanced workload with the **same** total FLOPs. Specifically, $B_i = 65536$ tokens are evenly distributed across $N$ experts, with $H = D = 8192$.

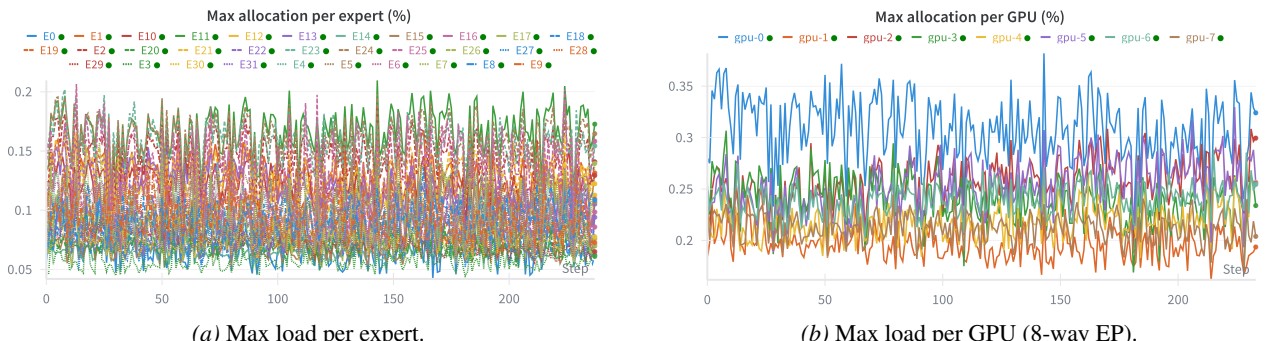

*(a)* Max load per expert.                    *(b)* Max load per GPU (8-way EP).

*Figure 9.* Expert routing imbalances across all layers of gpt-oss-20b across batches of a math dataset. **(a)** E11 has up to 20% load vs. ∼3% balanced. **(b)** GPU 0 has 30-35% vs. ∼12.5% balanced. Note that the load numbers do not add up to 100% because values are maximums across all layers.

means that at least one 11th expert of the 24 MoE layers consistently receives dominant load across data batches. Despite that, the GPU that E11 is located on, gpu-2, does not have the highest load; gpu-0, hosting experts E0-E3, has the highest load among all GPUs. This implies that certain GPU devices may handle an overwhelming number of tokens under extremely imbalanced routing. Interestingly, while E11 typically takes on the most tokens, certain batches result in more tokens routed to other experts. That is, the degree of imbalance changes on a per-batch basis.

## C.3. Number of Experts

Similar to the trends observe with batch size $B_i$ and hidden sizes $H, D$, Fig. 10 shows that LLEP is more efficient and exhibits greater speedups when the number of experts ($N$) in the MoE layer increases.

## C.4. Expert Load and Weight Allocations

To visualize how LLEP allocates loads and transfer expert weights, we build simple visualization tool that demonstrate how EP and LLEP distribute loads. Fig. 11 shows an example screenshot. Specifically, the first row shows the GPU load distribution across 8 GPUs. As seen, EP statically assigns loads only to the native GPUs, causing uneven distribution with GPU 6 having the highest load of 26K tokens. In contrast, LLEP dynamically balances loads across all GPUs, leading to the maximum load of ∼17K tokens. The second row shows more fine-grained details of load per expect. As seen, LLEP redirects portions of tokens of expert 3 (native GPU 0) to GPU 1, and expert 24 (native GPU 6) to both GPU 4 and GPU 7. The bottom table concretely shows the weight transfer between GPUs.

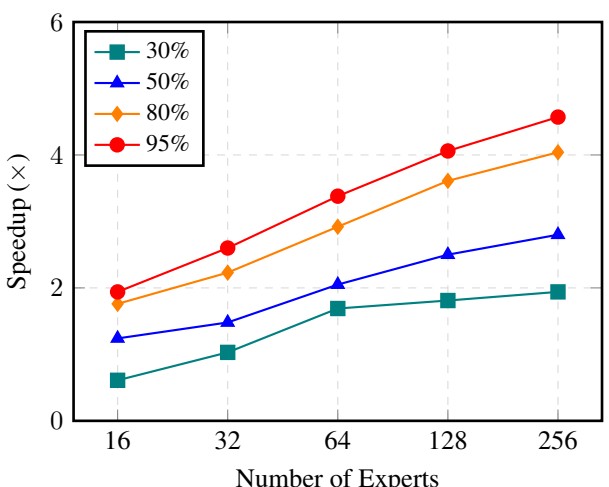

*Figure 10.* Speedup of LLEP over standard EP as a function of number of experts ($N$) with 4 imbalanced experts.

## 1. GPU Load Comparison 🔗

### Standard EP

Each GPU processes its native experts only.

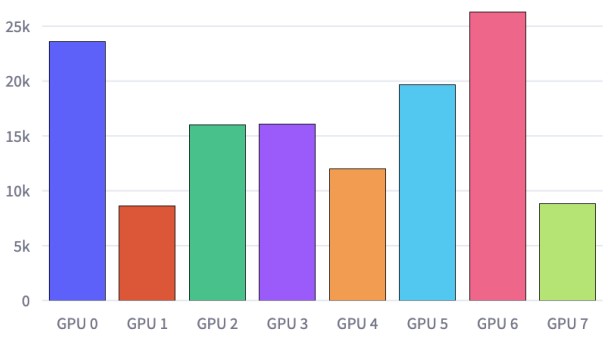

### LLEP / LLA (Solid=Native, Hatched=Spill)

Overloaded GPUs spill to least-loaded helpers, following LLAS rules.

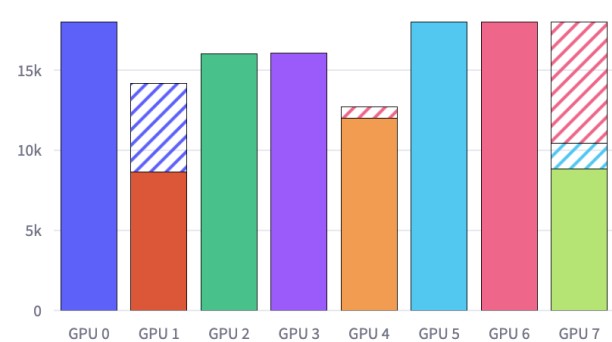

## 2. Experts' GPU Assignment

### Standard EP (Fixed)

Each expert is assigned to exactly one GPU.

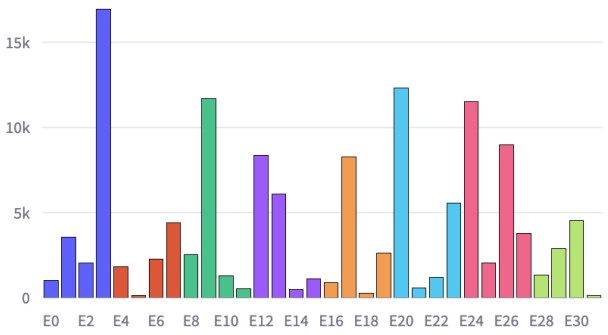

### LLEP (Split across GPUs)

Experts may be split across GPUs when spilling is needed.

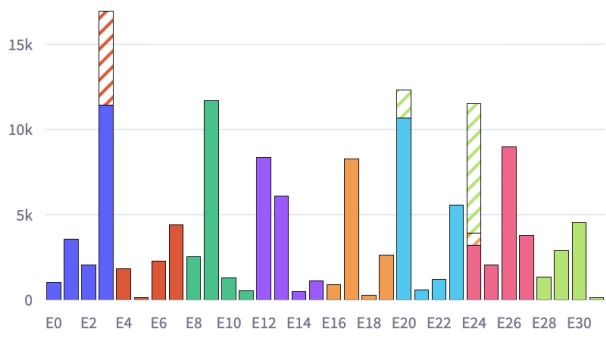

**Legend:** ■ GPU 0  ■ GPU 1  ■ GPU 2  ■ GPU 3  ■ GPU 4  ■ GPU 5  ■ GPU 6  ■ GPU 7

⌄  Show Plan Details

Weight Transfers Needed:  4

| expert_id | src_rank | dst_rank | token_start | token_end |
|---|---|---|---|---|
| 3 | 0 | 1 | 11417 | 16963 |
| 20 | 5 | 7 | 10683 | 12303 |
| 24 | 6 | 7 | 3202 | 10779 |
| 24 | 6 | 4 | 10779 | 11501 |

*Figure 11.* Example of Expert Load and Weight Allocations between EP and LLEP.