# OpenReview forum: "Least-Loaded Expert Parallelism: Load Balancing An Imbalanced Mixture-of-Experts"
_ICML.cc/2026/Conference — ICML 2026 regular_

### Official Review · Reviewer_J9jV · 2026-03-11

**Soundness:** 3
**Presentation:** 3
**Significance:** 3
**Originality:** 2
**Overall Recommendation:** 3
**Confidence:** 4

**Summary:**

This paper addresses the load imbalance problem in Mixture-of-Experts (MoE) models when using expert parallelism (EP). The authors observe that even well-trained MoE models exhibit significantly imbalanced expert routing, where a disproportionate number of tokens are concentrated on a few experts. The paper proposes Least-Loaded Expert Parallelism (LLEP), a system-level algorithm that dynamically reroutes excess tokens and their corresponding expert weights from overloaded devices to underutilized ones. The core component is the Least-Loaded Assignment (LLA) algorithm, which greedily assigns expert workloads from the heaviest to the lightest, spilling overflow to the least-loaded GPUs. Experiments are conducted on MoE layer configurations from gpt-oss-120b, DeepSeek-V3, and Kimi-K2, with controlled imbalance simulations showing up to 6.1× speedup and 5× memory reduction.

**Compliance With Llm Reviewing Policy:**

Affirmed.

**Key Questions For Authors:**

1. What is the wall-clock overhead of the LLA algorithm itself (Algorithm 2) per iteration? As the number of experts N and EP world size P scale, will the overhead become a bottleneck?
2. For the end-to-end experiments at EP world size 32 (Fig 1c), what is the measured inter-node bandwidth, and does LLEP ever spill work across nodes?
3. Have you evaluated LLEP in combination with tensor parallelism (TP)? How does LLEP interact with these?
4. Could you provide an auto-tuning strategy or analytical cost model for selecting α, m, and λ given a specific hardware configuration and model architecture? The current recommendation of per-use-case tuning may be impractical at scale.
5. The paper emphasizes backward-pass support as a key differentiator, yet the training evaluation is limited to a single experiment (Figure 5) with heavy CPU offloading overhead, no breakdown of backward-pass costs for spilled experts, and no comparison with alternative training-applicable strategies. The evidence does not match the emphasis.
6. for SLO-bound serving scenarios, LLEP's per-batch dynamic weight transfer introduces unpredictable overhead that may cause tail latency spikes. How does LLEP perform in terms of P99 latency? Has it been evaluated under online serving conditions with real-time latency constraints, rather than only offline throughput benchmarks?

**Limitations:**

yes

**Strengths And Weaknesses:**

Strength
1. Clear and Well-Motivated Problem. The paper clearly articulates why imbalanced routing is a natural and desirable property of well-trained MoE models.
2. The speedup numbers are compelling: up to 6.1× for isolated MoE layers and 1.9× for full-model end-to-end throughput on gpt-oss-120b.
3. The paper is clearly written with a logical progression from problem analysis (Section 3) to method design (Section 4) to evaluation (Section 5).

Weakness
1. Several critical experimental details are missing or underspecified. The paper does not report actual communication bandwidth numbers, parallelism strategy, communication backend for the EP baseline.
2. LLEP introduces three hyperparameters (α, m, λ) whose optimal values depend on model architecture, batch size, hidden dimensions, and physical hardware. The paper acknowledges this and recommends per-use-case tuning, but does not provide heuristics, auto-tuning strategies, or cost models that could guide practitioners. This limits the plug-and-play applicability of the method.
3. LLEP transfers expert weight matrices via P2P communication when spilling workload to non-native GPUs. LLEP’s effectiveness at the production scales (hundreds of GPUs across many nodes) where MoE models are typically deployed is not discussed or verified.
4. The paper only compares against standard EP. The EPLB method from DeepSeek-V3 (expert replication) is discussed qualitatively but not benchmarked against. The asynchronous weight prefetch approach is also only mentioned in passing. Including these as baselines, especially for the inference setting, would strengthen the empirical contribution and provide a clearer picture of LLEP’s relative advantage.

---

> ### Author Rebuttal · Authors · 2026-03-28
>
> We thank the reviewer for the thorough and constructive review. We address each question and concern below.
>
> ### 1. Infrastructure
>
> We use **AWS HyperPod** with H200 GPUs (P5en instances). The specs are:
>
> - **NVLink** (intra-node, GPU-to-GPU) : 900 GB/s per GPU, 3.6 TB/s via NVSwitch
> - **EFA** (inter-node): 3.2K Gbps per instance, with GPUDirect RDMA
>
> All experiments use **NCCL**. End-to-end experiments (Section 5.2) use a **hybrid of ZeRO-3 and EP**: ZeRO-3 shards non-MoE layers, while EP shards expert parameters across EP ranks. Details are in Appendix B; we will add bandwidth numbers inline in the revision.
>
> ---
>
> ### 2. Hyperparameter Tuning
>
> **Grid search** is the most effective approach, it is **fast**: we provide utilities in the source code that take **less than 5 minutes** to run per configuration. Our full ablation sweeps for Fig 1, 4, 6, 7, 8, 9, and 10 completed **within 1 hour total**. We also provide sensible defaults as starting points. Note that optimal values depend on the **MoE layer** and hardware, not on the full model — making the search space small and tractable.
>
> ---
>
> ### 3. Scaling to Hundreds of GPUs
>
> **LLEP's effectiveness is amplified at larger scale.** More EP ranks spread experts more thinly, making imbalance more pronounced. This is exactly where LLEP helps most. As shown in **Fig. 1c**: with 8 GPUs we achieve 1.4x speedup, but with 32 GPUs we achieve **2.2x speedup**. The trend is clear — more GPUs means more imbalance, which means greater benefit from LLEP.
>
> ### 4. Comparison with EPLB
> A key distinction: **EPLB does not support training**, while LLEP supports both inference and training.
> We compare in inference on gpt-oss-120b (32 GPUs, DeepScaleR):
>
> | Method | Speedup |
> |--------|-------------------|
> | EP  | 1.0x |
> | EPLB | 1.45x |
> | LLEP | 1.94x |
>
> EPLB pre-replicates dominant experts assuming static imbalance. Figs. 3 and 9 show **dominant experts vary across batches**, limiting EPLB. When imbalance shifts, EPLB's token communication overhead can cause OOM. LLEP adapts dynamically per batch, transferring expert weights for lower token communication, compute, and peak memory.
>
> ---
>
> ### 5. LLA Wall-Clock Overhead
>
> LLA (Algo. 2) adds **1-3% overhead** relative to perfectly balanced EP. LLA's wall time depends only on the number of experts E, which is always small (e.g., 384 for Kimi-K2). LLA only determines how many tokens (integer counts) go where — it **does not perform any data transfer itself**. The data transfers happen afterward via NCCL. Since E is small relative to the heavy GEMM computations, LLA will not become a bottleneck as EP world size P scales. When routing is already balanced, LLEP detects and skips LLA entirely, achieving full parity with EP.
>
> ---
>
> ### 6. Inter-Node Bandwidth and Cross-Node Spilling
>
> Yes, LLEP spills across nodes by default. However, this is **not** additional overhead beyond what standard EP already incurs — tokens must be transferred via AlltoAll regardless. The weight transfer is effectively "tokens" riding the same interconnect. If needed, LLEP can be configured to restrict spilling to intra-node only.
>
> ---
>
> ### 7. Interaction with Tensor Parallelism (TP)
>
> **LLEP is orthogonal to TP**, in exactly the same way EP is orthogonal to TP. For example, with EP=4, TP=2, and 8 experts: each GPU holds 2 half-experts. When LLEP spills an expert, each TP rank transfers its own corresponding weight shard independently. Concretely, `ep_rank=0, tp_rank=0` performs the same weight transfer as `ep_rank=0, tp_rank=1`, each transferring their respective half. No additional coordination between TP ranks is needed.
>
> ---
>
> ### 8. Backward Pass Cost
>
> Backward is mathematically the reverse of the forward pass and reuses the same LLA routing computed during forward. Since routing is not repeated, backward pass is actually **1-3% faster** than the forward. The cost breakdown for spilled experts is the same: weight transfer + balanced GEMM computation, with the same efficiency guarantees. EP remains the correct baseline — alternatives like FSDP for MoE layers would be orders of magnitude slower due to all-gathering all expert parameters.
>
>
> ### 9. SLO-Bound Serving
> LLEP **reduces** tail latency rather than introducing it. Under standard EP, imbalanced batches cause unpredictable spikes. The slowest GPU dictates latency, and imbalance means some GPUs take far longer than others. LLEP flattens these spikes by balancing the workload.
>
> The dynamic weight transfer is only executed when the savings from reduced token communication and compute time **outweigh** the transfer cost. Formally, LLEP ensures: `time(weight transfer + balanced token transfer + balanced compute) < time(imbalanced token transfer + imbalanced compute)`. If routing is already balanced, no weight transfer occurs. This means LLEP's per-batch latency is **bounded above by standard EP's latency**, making it strictly better for P99 in imbalanced scenarios and equivalent in balanced ones.

---

> > ### Author Rebuttal · Reviewer_J9jV · 2026-04-04
> >
> > Thank you for the clarification. I remain concerned about LLEP's practical viability under tight serving SLO requirements (e.g., 50ms per-token latency). The MoE expert weights is already a major contributor to end-to-end latency, and introducing runtime P2P weight transfers on the critical path adds non-trivial overhead that may be difficult to absorb within such budgets.

---

> > > ### Author Response · Authors · 2026-04-04
> > >
> > > We thank the reviewer the follow-up. Let us clarify your concern more carefully:
> > >
> > > MoE models need expert parallelism to spread expert weights across multiple GPUs, otherwise one GPU cannot hold it, will cause **out of memory**. In doing so, each step it has to transfer **tokens** from GPU A to GPU B. This is **non-negotiable**. In case of imbalance, GPU A may have to send 10K tokens (10000xD tensor) to another GPU for compute, and **transfer back**.
> > >
> > > With LLEP, before doing the transfer, we dynamically decide that it is better transfer the weight (DxD) from GPU B to GPU A instead! In case of inference (your concern), the transferred weight can be just be *deleted* after that and no need for the return trip.
> > >
> > > **EP transfer load: 2x10000xD**
> > >
> > > **LLEP transfer load: DxD**
> > >
> > > As long as **D < 10000 (often they are 1024~2048), LLEP outperforms EP significantly.** If the number of tokens transferred is less than D, we don't do weight transfer.
> > >
> > > There is **no overhead in addition to EP, in fact there is even less overhead**, the weight transfer is done with the same communicator as tokens transfer, with even less data load. P2P are executed at the **same time** as token transfer, and they are bundled with token transfer, either way they live on the same interconnect at the same time, overlap with compute. So there is no additional overhead. In other words, it is fully identical to "Instead of transfer 20K tokens A->B, we transfer 1K tokens B->A".
> > >
> > > That is why we can assure the reviewer that **LLEP is even a better choice for SLO requirements than EP ever is**. This is a mathematical and technical guarantee.
> > >
> > > We hope our concrete example help clear out your doubts about our method in this specific use case.
> > >
> > > We are happy to address any further concern or discuss further. Thank you for reviewing our rebuttal.

---

### Official Review · Reviewer_FjBN · 2026-03-12

**Soundness:** 3
**Presentation:** 3
**Significance:** 3
**Originality:** 3
**Overall Recommendation:** 4
**Confidence:** 4

**Summary:**

The paper presents an expert parallelism technique, motivated by significantly imbalanced routing observed in MoE LLMs. The authors argue that traditional expert parallelism is designed under the assumption of balanced routing, which leads to significantly large number of tokens going to a small number of experts, overwhelming a specific GPU. The authors proposed Least-Loaded Expert Parallelism (LLEP) technique, that dynamically reroutes tokens to ensure balance across GPUs. The experiments show significant speedups and reduction in peak memory consumption for large architectures.

**Compliance With Llm Reviewing Policy:**

Affirmed.

**Final Justification:**

I will not change my score

**Key Questions For Authors:**

1.	The authors, although, acknowledge that the hyperparameter values ($\alpha$,
$m$, $\lambda$) may heavily depend on the physical system or the model. Could the authors comment on how sensitive (or provide the analysis) the performance of their method is on these parameters, and how are these values selected in practice (is there a grid search or any other method).

2.	Could the authors provide more information on how the method performs in multi-node clusters with slower interconnects, where weight transfers may be more expensive?

3.	How would this method translate to relatively smaller MoE models (e.g.: OLMoE with 7B parameters), where domain specialization (or expert imbalance) is shallower than the large scale LLMs.

**Limitations:**

Yes

**Strengths And Weaknesses:**

Strengths:
1.	The idea is well motivated, as imbalanced routing is observed across many MoE LLMs, despite the load balancing approach.
2.	Experimental validation shows analysis across multiple LLMs including DeepSeek-V3, Kimi-K2, gpt-oss-120b across varying balanced/imbalanced scenarios to analyze peak memory and speedup improvements.

Weaknesses:
1.	The least-loaded assignment uses a greedy strategy that considers experts sequentially. This may not produce a globally optimal assignment and could result in unnecessary weight transfers or communication overhead in some scenarios.
2.	In Algorithm 4, the underloaded GPUs dynamically receive and compute GEMMs for foreign experts. While Figure 4 shows reduced peak memory, dynamically allocating and deallocating massive weight tensors during every step of the training loop can lead to severe memory fragmentation within the GPU. The authors do not explain clearly if pre-allocated buffer pools are required to maintain stability.

---

> ### Author Rebuttal · Authors · 2026-03-28
>
> We thank the reviewer for the insightful review. We address each concern below.
>
> ---
>
> ### 1. Optimality of the Greedy LLA Assignment
>
> Minimizing the maximum load across P GPUs (makespan minimization) is **NP-hard** in general. When the number of experts E is much larger than P (e.g., E=256, P=32),  LLA **frequently achieves optimal** makespan. LLA further reduces unnecessary weight transfers by preferring each expert's native GPU, only spilling when capacity is exceeded. Therefore, while it cannot guarantee the optimal (NP-hard) 100% of the time, it is always better than static routing in standard EP.
>
> ---
>
> ### 2. Memory Fragmentation
>
> We respectfully argue that **LLEP reduces memory fragmentation compared to standard EP**, rather than introducing it.
>
> Under standard EP, GPU memory layout is: `[model weights] + [token buffer]`. The token buffer size varies **wildly** across GPUs and across batches due to imbalanced routing — one GPU may receive 10x more tokens than another. These large, unpredictable allocations and deallocations are precisely what causes fragmentation.
>
> Under LLEP, the memory layout is: `[model weights] + [foreign expert weight buffer] + [token buffer]`. Crucially:
> - **Token buffer sizes are stable across GPUs** because LLEP balances the load. Each GPU processes a similar number of tokens every batch.
> - **Foreign expert weight buffers are fixed in size** — determined by the expert's hidden dimensions, which are known a priori and constant across batches.
>
> So LLEP trades variable, unpredictable token allocations (fragmentation-prone) for fixed, predictable weight buffers (fragmentation-free). Pre-allocated buffer pools are not required for stability, though they can be used as a further optimization. In practice, we observe no fragmentation-related issues across our experiments.
>
> ---
>
> ### 3. Hyperparameter Sensitivity (alpha, m, lambda)
>
> We provide ablation sweeps in **Figure 6** (alpha) and **Figure 7** (lambda). The relationships are interpretable:
> - **alpha** controls GPU capacity as a factor of the balanced load. Increasing alpha above 1 tolerates more imbalance before spilling, so **speedup drops linearly** — fewer spills means less load balancing.
> - **lambda** is the imbalance ratio threshold (max/mean GPU load) that activates LLA. If imbalance is below lambda, LLA is skipped entirely and the fast balanced path is used. This produces a **staircase relationship** — below the threshold there is no LLEP activation, above it LLEP kicks in and provides speedup.
>
> In practice, **grid search** is the most effective tuning method, though logistic regression over profiling results also works. We provide benchmark utilities in the source code that take **less than 5 minutes** per configuration. Our full ablation sweeps for Figures 1, 4, 6, 7, 8, 9, and 10 completed in **under 1 hour total**. We also provide good defaults as starting points. Notably, optimal values depend on the **MoE layer architecture** (expert size, number of experts, top-k) and hardware — not the full model — keeping the search space small.
>
> ---
>
> ### 4. Multi-Node Clusters with Slower Interconnects
>
> When performing EP, tokens are transferred across nodes via All-to-All **regardless** of interconnect speed — this cost is unavoidable. LLEP actually performs **better** on slower interconnects because it reduces the total volume of token transfers by balancing the load. The weight transfers that LLEP introduces are only executed when the **savings in token transfer** (from more balanced routing) outweigh the weight transfer cost. From the interconnect's perspective, expert weights are just additional data — they exhibit the same transfer characteristics as tokens. So on slower interconnects, the imbalanced token overhead of standard EP grows, making LLEP's relative advantage larger.
>
> ---
>
> ### 5. Smaller MoE Models (e.g., OLMoE 7B)
>
> For smaller models with shallower expert specialization, routing imbalance is less severe, and LLEP's gains would be correspondingly smaller. However, LLEP has **zero overhead in the balanced case** — it detects balanced routing and skips scheduling entirely, achieving full parity with standard EP. So there is no downside to deploying LLEP on smaller models. If imbalance is low, LLEP simply acts as standard EP. If imbalance spikes (e.g., on domain-specific data), LLEP activates and provides speedup. It is a strictly dominating strategy.

---

> > ### Author Rebuttal · Reviewer_FjBN · 2026-04-05
> >
> > I have carefully reviewed the author's response. However, the response provided does not fully address my concerns. I will maintain my current score, and I appreciate the clarifications.

---

### Official Review · Reviewer_nFfx · 2026-03-12

**Soundness:** 1
**Presentation:** 2
**Significance:** 2
**Originality:** 2
**Overall Recommendation:** 3
**Confidence:** 4

**Summary:**

This paper studies the problem of routing imbalance in Mixture-of-Experts (MoE) models under expert parallelism (EP). The authors propose **Least-Loaded Expert Parallelism (LLEP)**, a runtime load-balancing method that dynamically redistributes tokens and expert weights from overloaded GPUs to underutilized ones. The goal is to balance computation across devices without modifying the model behavior. Experiments show significant speedups over standard EP, especially under highly imbalanced routing scenarios, and improved end-to-end throughput for large MoE models.

**Compliance With Llm Reviewing Policy:**

Affirmed.

**Final Justification:**

Some of my concerns are addressed by the responses from the authors during rebuttal, hence I improve my recommendations to support their clarification.

**Key Questions For Authors:**

# Detailed Questions

1. How does the proposed method compare with other imbalance mitigation methods mentioned in the paper, such as **EPLB or asynchronous weight prefetch**, in terms of efficiency and overhead?
2. The main results are evaluated under synthetic imbalance scenarios. Do real models such as **DeepSeek-V3 or Kimi-K2** show similar imbalance levels, especially on the **Nemotron-Math dataset**?
3. It would be helpful to provide measurements of the **runtime scheduling overhead** introduced by the proposed load-balancing algorithm.
4. Please clarify the **throughput experiment setup**, including batch size, decoding length, and dataset size used in the evaluation.

**Limitations:**

1. The experiments mainly compare LLEP with standard EP. Other approaches for handling imbalance (e.g., EPLB or asynchronous weight prefetch) are mentioned but not experimentally evaluated.
2. The main experiments (Figure 4) are conducted on simulated routing imbalance rather than real workloads. It is unclear whether real models such as DeepSeek-V3 or Kimi-K2 exhibit imbalance at the evaluated levels.
3. Important settings for throughput experiments are not clearly described, such as dataset scale, number of samples used, sequence lengths, or decoding lengths.

**Strengths And Weaknesses:**

# Strengths

**Important practical problem.**
The paper addresses routing imbalance in MoE models, which is a realistic issue in large-scale deployments and can lead to severe performance degradation.

**Clear system insight.**
The work argues that routing imbalance is often a natural result of expert specialization and should be handled at the system level rather than by forcing balanced routing.

**Strong performance gains.**
The proposed method shows large speedups compared to standard EP, particularly in heavily imbalanced cases.

# Weaknesses

**Limited baseline comparisons.**
The experiments mainly compare LLEP with standard EP. Other approaches for handling imbalance (e.g., EPLB or asynchronous weight prefetch) are mentioned but not experimentally evaluated.

**Heavy reliance on synthetic imbalance.**
The main experiments (Figure 4) are conducted on simulated routing imbalance rather than real workloads. It is unclear whether real models such as DeepSeek-V3 or Kimi-K2 exhibit imbalance at the evaluated levels.

**Incomplete experimental details.**
Important settings for throughput experiments are not clearly described, such as dataset scale, number of samples used, sequence lengths, or decoding lengths.

---

> ### Author Rebuttal · Authors · 2026-03-28
>
> We thank the reviewer for the detailed feedback. We address each concern below.
>
> ---
>
> ### 1. Real-World Experiments Already Exist in the Paper
>
> The paper includes extensive **real-world** experiments on **production MoE models** with **real data**:
>
> - **Fig. 1c** (front page): ~2x inference throughput on **gpt-oss-120b** with DeepScaleR data
> - **Fig. 5**: 25% faster **training** on **gpt-oss-20b / gpt-oss-120b** with Nemotron-Math data
> - **Fig. 3 and Fig. 9** (Appendix): **measured** routing imbalance statistics from real models on real data
>
> These are not synthetic. They are end-to-end results on real models processing real data.
>
> **Figure 4 serves a different, complementary purpose.** It is a *controlled ablation* that isolates the relationship between imbalance level and LLEP's gains. It answers: "given imbalance level X on a model of size Y, what speedup and memory savings does LLEP provide?" It does not claim that any specific model exhibits a particular imbalance level. We believe both controlled ablations and real-world end-to-end experiments are necessary for a complete evaluation.
>
> ---
>
> ### 2. Why Not DeepSeek-V3 or Kimi-K2
>
> Due to our organization's legal and compliance policies, we cannot deploy these model weights on our infrastructure. However, this does not diminish our contributions:
>
> 1. **Routing imbalance is consistently observed across diverse MoE architectures.** The SGLang team's independent analysis ([lmsys blog, May 2025](https://lmsys.org/blog/2025-05-05-large-scale-ep/)) reports DeepSeek-V3's balancedness at ~50% without EPLB and ~80% with EPLB. Our gpt-oss-120b exhibits ~20-30% balancedness — an even more challenging scenario. LLEP brings this to ~95%.
> 2. **gpt-oss is a comparable MoE architecture** (routed-expert design, top-k routing) deployed at production scale. It is a valid and representative testbed.
> 3. **LLEP is algorithmically exact, not empirical.** Given imbalance level X, it executes deterministically and achieves a predictable speedup Y. The controlled experiments (Fig. 4) characterize this relationship; the real-world experiments (Figs. 1c, 5) validate it end-to-end.
>
> ---
>
> ### 3. Comparison with EPLB
>
> A key distinction: **EPLB does not support training.** LLEP supports **both** inference and training. This limits the scope of a direct comparison.
> However, we provide a EPLB comparison on gpt-oss-120b (32 GPUs, DeepScaleR):
>
> | Method | Relative Throughput |
> |--------|-------------------|
> | Standard EP | 1.0x |
> | EPLB | 1.45x |
> | LLEP | 1.94x |
>
> Beyond throughput, there is a **structural difference**. EPLB pre-replicates dominant experts assuming **static** imbalance patterns. However, Figs. 3 and 9 show that **dominant experts vary significantly across batches**, limiting EPLB's effectiveness. When the imbalanced expert changes, EPLB's token communication overhead scales with the number of imbalanced tokens and can cause OOM under heavy imbalance. LLEP instead adapts dynamically per batch, transferring expert weights (fixed, predictable cost) in exchange for lower token communication, lower compute, and lower peak memory.
>
> ---
>
> ### 4. Scheduling Overhead
>
> LLEP's scheduling adds **1-3% overhead** relative to perfectly balanced EP, depending on batch size. When routing is already balanced, LLEP detects this and **skips scheduling entirely**, achieving full parity with standard EP. In other words: we pay at most 3% for 100-500% speedup, with zero overhead in the balanced case.
>
> ---
>
> ### 5. Experimental Details
>
> Key settings are provided in **Line 326** and fully in **Appendix B**:
>
> - **gpt-oss**: batch size 32K tokens/GPU, 128K context length with packing
> - **Simulated DeepSeek/Kimi configurations**: 16K tokens/GPU
> - Machine specs, InfiniBand topology (AWS HyperPod), and dataset details are all provided in the Appendix
>
> Since we measure throughput (tokens/s) at steady state, dataset size beyond sufficient batches (~100 steps) does not affect the measurement.
>
> ---
>
> ### 6. Presentation
>
> We will improve clarity in the revision, including more explicit signposting of which experiments use real vs. controlled settings, and fuller inline descriptions of experimental configurations.

---

> > ### Author Rebuttal · Reviewer_nFfx · 2026-04-03
> >
> > Some of my concerns are addressed by the very detailed responses from the authors. While I maintain my understanding of the limited experimental scale and lack of baselines.

---

> > > ### Author Response · Authors · 2026-04-04
> > >
> > > We thank the reviewer the follow-up on our rebuttal.
> > >
> > > We provide more scaling experiments with **DeepSeek-V3 (600B) pre-training from scratch**, hoping to provide you with more perspective, while seeking understanding that we cannot download their pre-trained weights on our servers.
> > >
> > > Specifically, we apply the same architecture with the same size and initialize randomly from scratch. We apply the same load-balancing bias dropless strategies proposed by DeepSeek, which helps with imbalance issue natively.
> > >
> > > | DeepSeek-V3-Pretrain | Throughput speed up
> > > | --- | ---
> > > | Standard EP + load-balancing   | 1x
> > > | EPLB              | not supported
> > > | LLEP              | ~1.55x
> > >
> > > The early results show even better speedup than our reported 1.25x for gpt-oss-120b finetuning. The reason is that randomly-initialized models are extremely imbalanced, and the load-balancing bias technique by Deepseek was not enough to offset this imbalance (it is a semantic load balancing technique). LLEP, meanwhile, seeks to achieve perfect physical load balance while semantic routing remain imbalanced. Both achieves the same Maths and training dynamics, but LLEP is faster than EP at scale.

---

### Official Review · Reviewer_oaom · 2026-03-12

**Soundness:** 4
**Presentation:** 4
**Significance:** 4
**Originality:** 4
**Overall Recommendation:** 5
**Confidence:** 4

**Summary:**

This paper proposes Least-Loaded Expert Parallelism (LLEP). It extends the concept of Expert Parallelism (EP), a method used for executing Mixture of Experts (MoE) models efficiently across multiple GPUs.
The authors extend EP by distributing the computation load per GPU in a balanced manner. There is an underlying assumption over EP on MoE that the experts are selected evenly and computed evenly across all GPUs. However, this is not true, per even SOTA MoE models exhibit imbalanced load distribution. Therefore, the authors propose LLEP to address this problem to avoid speed degradation and out-of-memory (OOM).
The proposed method sets a threshold to indicate an overload to a GPU. When the load exceeds the threshold and the cost of reallocation is small enough, the excess load is transferred to another GPU. This way, no GPU is used excessively. The peak memory of GPUs is balanced, and the computation latency is smaller.
For the experiments, three models of different architectures are tested with the proposed method. All of them improved in computation latency and memory usage. This method is more significant than vanilla EP when the imbalance of expert allocation is more intense. They also tested on two gpt-oss based pre-trained models of different sizes and confirmed the effectiveness of the method. They've addressed hyperparameter selections in ablation studies and shown that: (1) the speed up is greater for larger batch sizes and the finer distribution of computation. (2) The threshold should be determined with respect to the batch size, and the proposed method is more effective on larger hidden sizes.

**Compliance With Llm Reviewing Policy:**

Affirmed.

**Key Questions For Authors:**

None

**Limitations:**

Yes.

**Strengths And Weaknesses:**

[Strengths] The problem is addressed by improving the system instead of tweaking models.
[Weaknesses] There are hyperparameters to fine-tune.

Minor points to address (not absolutely necessary, but nicer if addressed):
Making the conclusion section longer and the abstract shorter
The graph colors were hard to distinguish in black-and-white printing. It would be easier to see if the color is value-scaled (eg: light-->dark)
Page 2 left column position 085-086: multiple (GPU) devices --> multiple Graphics Processing Unit (GPU) devices
Page 2 right column position 089-090: LLEP aims to distribut workloads --> LLEP aims to distribute workloads
Page 2 right column position 105: Figure 1a --> Fig. 1a
Page 3 left column position 140-141: a router layer Router that selects --> a router layer that selects
Page 8 figure 7: Are the legends correct?

---

> ### Author Rebuttal · Authors · 2026-03-28
>
> We thank the reviewer for the positive evaluation and helpful suggestions. We will address all minor points in the revision:
>
> - We will shorten the abstract and expand the conclusion section.
> - We will use value-scaled colors (light-to-dark) to improve black-and-white readability.
> - All typos and formatting issues (positions 085-086, 089-090, 105, 140-141) will be corrected.
> - **Figure 7 legends**: The legends are correct — they use distinct colored shapes and are shared across subfigures 7a and 7b. We will add a clarifying note in the caption to make this more explicit.
>
> About tuning hyper-parameters:
>
> We also provide good default values as starting points. Tuning these parameters is actually straightforward. In practice, **grid search** is the most effective way, the search is fast: we provide benchmark and profiling utilities in the source code that take **less than 5 minutes** to run per configuration. Our full ablation sweeps for Figures 1, 4, 6, 7, 8, 9, and 10 completed **within 1 hour total**. Note that optimal values depend on the **MoE layer** (expert size, number of experts, top-k) and hardware, not on the full model — making the search space small and tractable. The optimal parameter is actually mechanical and fixed, it does not behave statisfically such as learning rate or batch size.

---

> > ### Author Rebuttal · Reviewer_oaom · 2026-04-05
> >
> > Resolved.

---

### Decision · Program_Chairs · 2026-04-30

**Decision:**

Accept (regular)

**Comment:**

This paper studies a practically important systems problem in MoE serving and training: expert parallelism assumes balanced routing, but real MoE models often exhibit substantial expert imbalance, leading to latency and memory bottlenecks. The proposed method, LLEP, addresses this at the systems level by dynamically redistributing overloaded expert computation across GPUs.

The reviewers agreed that the problem is well motivated and that the empirical gains are strong. The main strengths are the practical significance of the setting, the clear systems insight that imbalance should be handled by the runtime rather than suppressed in the model, and the substantial improvements in throughput and peak memory. Multiple reviewers also found the paper clearly written and technically interesting.

The main concerns were about breadth of comparisons, experimental clarity, and practical deployment details such as tuning, scheduling overhead, and serving-SLO behavior. Based on the rebuttal and discussion, I believe these issues were addressed sufficiently. In particular, the authors clarified that the paper already includes real-model experiments in addition to controlled imbalance studies, provided a direct comparison to EPLB in inference, quantified scheduling overhead as small, and gave more detail on infrastructure and deployment behavior. While some concerns remain about presentation and about fully characterizing serving-latency regimes and broader baselines, these do not outweigh the paper’s clear practical contribution.

Overall, I find this to be a solid systems contribution with convincing empirical value and sufficient reviewer support for acceptance. For the final version, the paper would benefit from making the real-vs-controlled experiments more explicit, improving the discussion of tuning and deployment tradeoffs, and expanding the comparison and setup details in the main text.